# Design and evaluation of a problem-based learning VR module for apparel fit correction training

**Aditi Galada, Fatma Baytar** [ORCID] *

Department of Human Centered Design, Cornell University, Ithaca, New York, United States of America

* baytar@cornell.edu

## Abstract

The increased adoption of three-dimensional (3D) digital prototyping software programs makes it necessary to train novice designers to use these programs efficiently. However, existing studies spanning from engineering to design education indicate that students feel incompetent in understanding 3D digital prototypes and navigating the software, so there is a need to find effective training methods. In the current study, training modules were developed to teach participants fit correction skills through an iterative problem-based learning (PBL) approach. A review of the literature was performed to develop the fit correction tasks and guide the module development process. Expert feedback was used to fine-tune the tasks and module interface. The current study explored the effects of PBL-based virtual reality (VR) training on learning how to correct two-dimensional (2D) apparel patterns to improve consequent 3D garment fit. Results indicated that the training module significantly improved spatial visualization and fit correction skills. Participants with higher apparel spatial visualization skills saw a higher improvement in fit correction skills because of the training. At lower spatial visualization skill levels, women saw a higher increase in apparel spatial visualization skills after the training than men but the difference between the learning outcomes across genders reduced when participants had higher spatial skills before training. These findings were supported by the results of qualitative data obtained through interviews. The participants found the PBL approach, immediate feedback, and aid in visualization through garment simulations beneficial for understanding the concepts. At the same time, they indicated that the training module could be used as a supplement to the traditional classroom but cannot replace the physical garment fitting practice. The findings of the study verified that PBL using virtual garment simulations can have a positive impact on learning outcomes and help identify the stage of education at which learners can be exposed to PBL.

## Introduction

The apparel patternmaking process is an iterative trial-and-error process of identifying fit issues and correcting them on the test garments, or prototypes, to ensure patterns to be cut

**Data Availability Statement:** All data, code, and log files are available from the Cornell eCommons data archive. The files can be accessed by following this DOI link: https://doi.org/10.7298/8wsv-0q41.

**Funding:** This material is based upon work partially supported by the National Science Foundation under Grant No. 2048022 to FB. Any opinions, findings, and conclusions or recommendations expressed in this material are those of the authors and do not necessarily reflect the views of the National Science Foundation. (http://www.nsf.gov/) The funders had no role in study design, data collection and analysis, decision to publish, or preparation of the manuscript.

**Competing interests:** The authors have declared that no competing interests exist.

during mass production would result in garments that would fit well. In the traditional patternmaking process, each physical prototype revision requires time and adds to the expenses. Therefore, the supplementation of traditional fit analysis sessions with virtual garment simulation on scanned fit models is gaining traction. Companies including CLO [1], Browzwear [2], and Optitex [3] allow designers to create garment simulations from 2D patterns on custom avatars. These software programs help visualize the designs, detect errors, and rectify them in the early stages of prototyping by enabling designers to view the effects of 2D pattern modifications on the 3D garment fit in a matter of seconds. As a result, 3D prototyping provides the advantage of reduced lead time and decreased wastage by limiting the number of physical prototypes [4]. When using virtual fit models, the garment fit can be tested for several sizes and on models with different body shapes. By maximizing the potential of fit sessions, brands can improve customer satisfaction regarding garment fit and, as a result, increase sales [5].

Industry professionals have indicated that design students should have 3D garment design skills [6]. Graduates applying for technical designer positions are expected to be proficient in using 2D/3D CAD software programs to maximize efficiency and minimize the cost of producing garments [7]. However, apparel fitting is a complex process because the relationship between the body and the garment is often ambiguous. In academia, students are often trained to fit garments on dress forms with hourglass body shapes. Without learning how to identify and resolve garment fit issues on various body forms, and consequently, manipulate the 2D patterns to fit flawlessly, new graduates or novice designers will have lower chances of being successful in the industry [8]. It can take over a decade of experience for novice designers to identify and rectify fit issues accurately [9, 10]. This knowledge gap creates a difference in the way people understand and interpret sensory information [11]. Experts recognize patterns, predict problems, and make better decisions based on the same sensory information [12].

VR provides a fun, interesting, and immersive way of learning [13]. Integration of garment simulations in a PBL module can allow higher interactivity as well as control over the animations. PBL provides learners with a task that closely resembles a real-world problem and allows them to learn by doing. Several studies have been performed on the effectiveness of PBL, but they are frequently related to Science, Technology, Engineering, and Math (STEM) education. The PBL education in the apparel design domain remains to be explored. Using multi-media technologies to develop PBL tasks has been identified as a solution to impart critical and analytical thinking to students [14]. VR can provide a platform to examine and modify prototypes in a real-world setting [15].

Past research mainly focused on developing the technology to enable 3D visualizations and interactions in the field of fashion design [16] but there is a growing need to examine how to use this technology more effectively in the field of apparel design education. Therefore, the main aim of the present study was to examine how VR could be used to reduce the gap between the knowledge of experts and novice designers by using interactive virtual garment simulations in PBL modules on conceptual fit understanding and spatial skills. The study evaluated PBL by using a desktop VR (d-VR) training module to examine VR's usefulness in providing fit correction experience, which is usually not possible without years of work, to designers. By using a one-group pretest-posttest design, the objectives of the present study were four-fold: (1) to develop PBL-VR training modules for garment fit correction, (2) to evaluate if PBL-VR modules can improve (a) spatial visualization skills and (b) fit correction skills, (3) to analyze if pre-training apparel spatial visualization skill level would have a relationship with fit correction skill improvement after a PBL-infused VR training and (4) to examine if gender plays a role in performance improvement through PBL-VR modules.

## Literature review

### Apparel fit analysis

Apparel fit is the relationship of clothing to the contours, shape, proportions, and size of the human body [17]. Fit has three aspects: (1) aesthetic fit refers to self-evaluation of the overall appearance of the garment, (2) functional fit which depends on the level of comfort the wearer feels when wearing the garment, and (3) physical fit is determined from the tightness and length [18]. Fit issues in garments often arise from differences in body shape [19, 20]. As a result, the fitting process is different for everyone as no two bodies are the same. A standard pattern with the measurements closest to body measurements is usually selected and altered to fit the body correctly. The patterns need to be modified as measurements alone do not reflect body shape [21].

Elements of fit that must be checked to ensure the garment is comfortable and conforms to the body include ease, line, grain, balance, and set [22]. Ease is the space between the garment and the body and can be divided into functional and design ease. The line refers to the alignment of seams. Vertical seams must be perpendicular to the floor. Grain represents the relationship between the direction of the fabric and the alignment of the pattern on the wearer's body. Balance refers to a garment that hangs on the body evenly on the right and left sides. Set indicates the absence of pulls, folds, and wrinkling [22].

Pattern modification procedures differ for people with different body shapes. Therefore, it is essential to train designers in making the right pattern alterations. Experts utilize the knowledge gained through experience to understand body-to-pattern relationships for individual customers with unique body shapes and sizes. Each fitting process has a unique interaction between measurements, shape, ease, fabric, and personal preference; this information cannot be conveyed through texts, instead, designers develop an intuition through experience [8, 21].

In previous study, a standard framework to evaluate fit based on ease, line, grain, balance, and set was developed [23]. The participants indicated if there were many to no wrinkles/ folds/ gapping etc. on a 4-point Likert scale. The researchers compared the results of the subjective traditional expert fit analysis to that of image analysis and found that image analysis was a valid objective measure of garment fit. In another study, researchers recruited students and expert judges to analyze fit [24]. The students were trained using training modules and handouts. The training modules consisted of video recording of a model wearing different jackets with unique fit issues. The recording explained the visual cues that represent a good fit. Additionally, a handout was given that included fit principles and a checklist. To evaluate students' and expert judges' ability to assess garment fit, recordings of women's dresses in the commercial jacket were shown, and a fit checklist was handed out to collect data. The results of the study indicated that the training can be used to match the fit assessment skill of inexperienced designers to the fit assessment skill of experienced judges.

**Fit analysis using 3D simulations.**   With the improvements in computer graphics and due to the manufacturing requirements, such as faster lead times, the use of 3D simulations in the apparel industry has been increasing [25]. There is a gap between the 3D skills required in the industry, therefore, training and education systems need to be updated to promote the digital competencies of employees [26]. Apparel 3D prototyping software programs support designers in identifying issues related to fit or design without creating physical samples and often achieve a close-to-real visualization of the actual garment silhouette [27]. Users can rotate the model and zoom in, which helps visualize how the actual garment would look [27]. The software programs provide an opportunity to check various patterns, colors, fabrics, etc. to see the change in appearance and drape of the clothing [25]. Additionally, virtual avatars

can be 3D body scans of real people, or fit models. Testing fit on custom avatars enables remote fit sessions, testing on multiple models, and repeat evaluations [28].

There is an advantage of providing the instantaneous effects of 2D pattern changes in 3D [29]. Multi-media tools such as 3D simulations are known to reduce cognitive load and increase learning gain and engagement [29, 30]. Training using 3D simulations can provide technical advantages related to interconnected 2D and 3D visual aids, enrich content through the development of interactive modules, and promote learning by providing the ability to view the realistic clothing simulation from multiple angles. As a result, the time required to train users in 3D patternmaking methods can be less than in 2D patternmaking methods [10].

A previous study compared learnings after each medium of instruction, including lectures, paper pattern making, and 3D simulation [31]. The researchers employed a notch-matching exercise to evaluate the improvement in visualization skills. The ratings and scores on the post-training questionnaire showed that the 3D simulations escalated the learning curve. Another research study used pre- and post-training surveys to evaluate the improvement in the performance of students after repeated use of 2D/3D CAD software programs [29]. Results indicated an improvement in interaction, 3D visualization, and problem-solving skills of apparel design students which made it easier to use the CAD software. Recently, researchers also used pre-and post-tests and interviews to evaluate training modules developed for virtual technologies in the apparel industry [32, 33]. Results showed that the spatial visualization skills of students improved, and they developed a positive attitude toward the technology.

## Virtual reality (VR) as a facilitator of learning

Traditional learning methods are considered to lack the connection between theoretical knowledge and its application in a real-world scenario [34]. The work environment requires people to think critically and be technologically literate, but it is not possible to teach these skills using conventional methods. [35]. There are several theories developed to better understand and formulate how people learn, apply, and retain information. These in turn help adapt the educational material to different learning styles to enable the effective acquisition of knowledge and skills [36]. These theories include, but are not limited to, Experiential Learning, which consists of four stages related to concrete experience, reflective observation, abstract conceptualization, and active experimentation [37]; Constructivist Learning Theory, which is built on the basis that learners create their knowledge by analyzing new information and connecting it with their existing experience [38]; and PBL theory, which switches the traditional theory to problems approach into a bottom-up approach where the problem is followed by the theory [39].

In PBL, learners are provided with close to real-world problems as a trigger or starting point and gain knowledge through their experiences [40]. The steps in PBL include exploration of the problem, defining the problem, data gathering, analysis of possible solutions, making a decision, and reflection [41]. VR has been used as an effective medium in PBL to support experiential learning as it allows users to interact with objects in the virtual world [42] and provides an opportunity to learn through experience [43]. VR is defined as "a computer-mediated simulation that is three-dimensional, multisensory, and interactive so that the user's experience is "as if" inhabiting and acting within an external environment" [44, p.37]. As a result, learners store this knowledge in their memory which improves their performance in real-world scenarios [45]. Through its nature, VR could be an effective learning environment to improve 3D digital skills and learning.

There is a need to educate the current and future workforce with increased 3D skills in the apparel product development field [33]. There is also a growing need to develop skills in

"reading clues in 3D" and translating them into 2D patterns [31]. PBL modules can be powerful tools in helping students learn apparel fit correction through an interactive interface. To be able to do this, a feedback loop must be created to improve learning and guide the learner through the VR environment [46]. To improve decision-making skills in a high-pressure environment with changing conditions and extensive information, the 'recognition-primed decision model' is considered the most efficient [47]. According to the model, the decision maker generates an option, tests it for feasibility, and decides whether to implement or reject the option. Experienced employees are usually more proficient in using their cognitive ability to generate the relevant options, but novice employees find the process difficult [12]. Technology-mediated learning where users can interact with objects proves to be helpful when training learners on how to use the model [12]. The ability to interact with objects in 3D improves the effectiveness of training [48]. Additionally, providing immediate feedback and assistance in visualizing concepts the VR platform helps build skills and capabilities [49]. The design of the training module in terms of navigation and interaction should be given utmost importance as it has a large impact on learning effectiveness [50].

The visualizations and simulations provided by garment simulations can enable PBL through active learning. In domains where the spatial and visual aspects carry importance, VR can provide additional advantages [51]. Apparel spatial visualization skill refers to the ability to transform 2D patterns into 3D garments through mental rotations, imagining the folding and unfolding of patterns, changing the relative position of patterns, and so on [52]. Past research indicates that apparel spatial visualization skill improves as an indirect effect of apparel design training [53]. Considering the findings from past research, the following research question (RQ) was formulated to evaluate the impact of the novel PBL-VR training module that was specific to the field of apparel design:

RQ1: Can PBL-VR training increase (a) apparel fit correction skills and (b) apparel spatial visualization skills?

## Individual differences in technology-infused learning performance

**Prior experiences/ skills.** Prior knowledge supports learners to pay attention to the training content as well as aid in visual processing [54]. The spatial visualization skill plays a crucial role when determining success when learning through VR [55]. Learners with higher spatial abilities learn more when training through VR [55]. Therefore, apparel spatial visualization skills can be expected to play a crucial role in predicting the learning outcomes as garments are complex 3D objects. Students can apply their prior knowledge, or existing skills to make the right decisions in apparel fit corrections. The following RQ explores the relationship between prior apparel spatial visualization skills on the benefits gained from the PBL-VR training:

RQ2: Would the people with higher apparel spatial visualization skills experience a larger improvement in their fit correction skills after the PBL-VR training?

**Gender.** A difference in performance based on gender has been noted by academic research. While some studies indicate that women outperform men in terms of learning outcomes [56], other research studies indicate that men perform better than women in VR [57], and some studies indicate that gender has no difference in performance [58, 59]. These differences might exist because of differences in training module complexity as well as the context. Apparel design education has traditionally drawn more women than men. In 2021, it was reported that women predominate in the fashion designer position [60]. Therefore, in the apparel design education context, women can be expected to perform better than men in VR conditions, however, this notion has never been explored before. The following RQ was structured to examine this expectation:

RQ3: How does gender play a role in improving (a) apparel spatial visualization skills and (b) fit correction skills after a PBL-VR training?

The final research question was developed to analyze the perceptions toward the PBL-VR training:

RQ4: What perception would users have about the PBL-VR training?

## Methodology

### VR training module design and development

To develop training modules, previous literature on indicators of a good fit, recognizing fit issues and rectifying them, previous training modules developed to educate apparel design students, and the tools used to measure domain-specific learning were analyzed thoroughly. The training modules were designed to teach the participants to use the 'recognition-primed decision model' [47] when determining what pattern modification option would correct the garment fit. By following the Experiential Learning Theory, the modules were designed to provide concrete experiences on fit correction, allow learners to reflect on their choices based on the immediate audio-visual feedback, and generate new conceptualizations based on the explanations provided on how the pattern modification corrected the garment fit. With multiple unique tasks, the learners could apply the new knowledge to real-world tasks. As the Constructivist Learning Theory outlines, the users could explore different fit correction tasks, make decisions, and learn by observing the impact of their decisions. As recommended by the PBL theory, the users were provided with the context of the fit issue as well as guidance on how to solve the problem. To make sure the module was accessible to a broader audience the controls and navigation were planned to be easy to use. For novice-level designers, the PBL module should not be overwhelming by making it extremely close to reality [61]. Therefore, providing prompts and visual aids at each stage was considered important. To mitigate potential bias during the development stage, the team that developed the training module consisted of members from diverse backgrounds including game development, apparel design, and human computer interaction. This ensured that a single perspective did not dominate the development process. To reduce bias during evaluation of the training module, pre-tested assessment tools combined with blind evaluation were applied.

**Fit correction tasks.**  Five iterative problem-solving tasks, i.e., a series of pattern modifications to achieve a good fit, were adapted from [21]. The garments included a sleeveless top, slacks, a full-sleeve t-shirt, a sleeveless dress, and a skirt. Selected fit issues, the reasons for the fit issues, and the expected pattern corrections are listed in S1 Fig. The virtual garment simulations were made using Clo3D and exported as obj files.

**VR interface.**  The virtual garment obj files were imported into Unity where they were incorporated into the PBL-VR modules. The users viewed the modules through a monitor and used a mouse to navigate the modules and complete different fit correction tasks by clicking on the desired option. The development and implementation of the module were cost-effective, and the interface provided users with an easy-to-navigate environment through a semi-immersive experience. The training modules (Fig 1) included PBL learning activities where the user was given a brief explanation of the garment fit issue that needs to be rectified, shown the simulated garment with fit issues, and given three potential pattern modification options one of which would correct the fit issue. The users could interact with the model and view it from different angles to better understand the garment fit. If the user selected the wrong pattern modification option, immediate audio-visual feedback was given as seen in Fig 1(E),

which was expected to reinforce learning. In case the user chose the right pattern modification option, audio-visual feedback indicated that the right option had been chosen as shown in Fig 1(F). The user was then directed to the next step where the 3D garment simulations with the before and after the pattern modifications were shown along with the correct pattern modification as well as an explanation of why the pattern change corrected the fit as shown in Fig 1(G). Once all the fit issues in a garment had been rectified, the participants were shown their score and directed back to the main menu to work on the next garment.

**Pilot test.** A five-person fit expert panel evaluated the training modules before data collection. Experts answered the following questions: "Did you face issues with any particular question?", "What do you think about the user interface?", "Did you face any challenges navigating the training module?", and "Was the difficulty level of the pre-and post-test comparable?". They also provided their feedback on the prompts, answer choices, and interface through a survey distributed on Qualtrics containing open-ended questions. The issues identified related to the navigation, interaction, and the module in general as well as corrections related to specific fit correction questions are listed in Table 1 along with the applied solutions.

Overall, the expert panel appreciated step-by-step solutions, immediate feedback, descriptive text about the fit issue, and an explanation of how the pattern modification corrected the fit. Multiple experts mentioned that the repetitive tasks aid in learning, however, they emphasized that without an understanding of interpreting folds and draglines, novice designers might not understand how to deal with fit issues. Therefore, it was decided to show participants a short video on how to recognize and rectify fit issues before starting the PBL training module. The video explained how to identify the origin of the issue based on the draglines, folds, and wrinkles and their direction on the garment.

The experts suggested that adding a tutorial on how to navigate the training modules would be beneficial to learners. Therefore, one task was added as a tutorial that explained how to navigate the environment, interact with models, select the pattern modification answers, and what to expect when they chose the right/ wrong option. The experts also acknowledged the benefit of 3D simulations as they allowed users to analyze the garment from different angles which in turn helped gain visual and critical practice.

## Data collection

After receiving approval from the Institutional Review Board (protocol no.2108010511), participants were recruited from Cornell University. The focus on college students was intended to provide a targeted examination of the potential benefits of incorporating VR into the curriculum. Providing incentives such as monetary compensation as well as credit in courses for participation that appealed to a wide range of individuals and not just people interested in technology helped avoid selection bias. The recruitment period for the study was from August 29, 2022, to October 1, 2022. At the beginning of the session, the purpose of the study was explained to the participants orally. Participants were also informed with a written consent form, which they read and signed. The training modules were aimed at students as they represent the future workforce in apparel design. No pre-requisite knowledge was expected of the participants or viewers. A quasi-experimental, pretest posttest design was used. Before and after the training participants completed a questionnaire that consisted of demographics, prior pattern making experience, and two objective performance measures, i.e., Apparel Spatial Visualization Test (ASVT) questions adapted from [52] and the Fit Correction Skill Test (FCST) which was developed in a previous study [61]. While the FCST was designed in a 10-question each for the pre-and post-test format, the 20 ASVT questions were equally divided

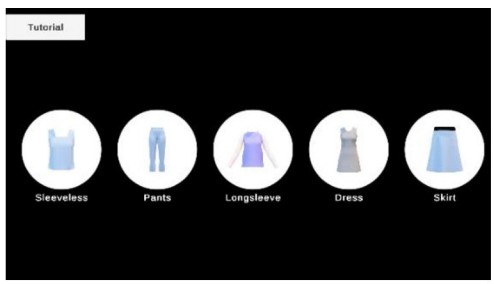

(a) Main Menu

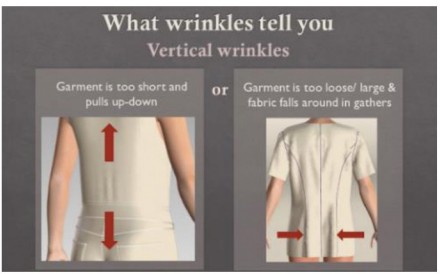

(b) Training Video

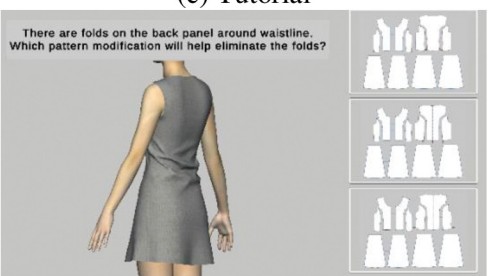

(c) Tutorial

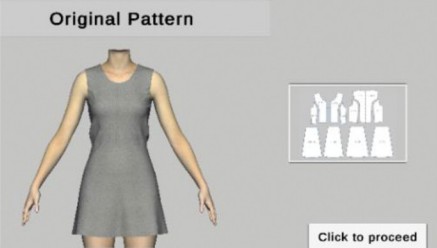

(d) Garment with fit issues

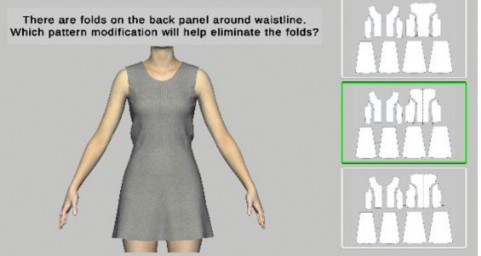

(d) Fit correction step with prompt and options

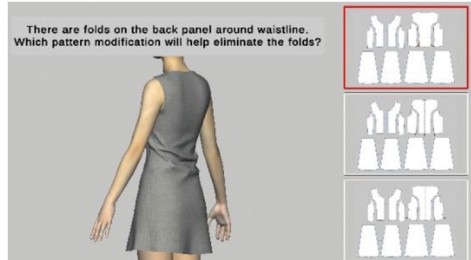

(e) Feedback on selecting the wrong option

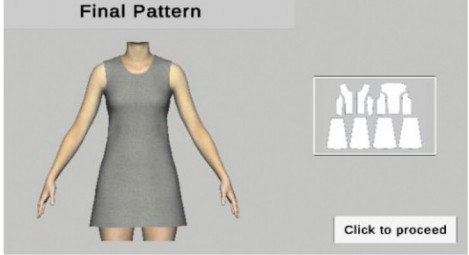

(f) Feedback on selecting the right option

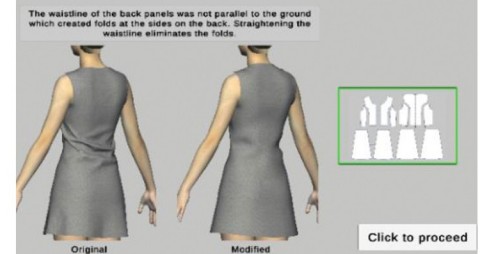

(g) Explanation of how the fit was corrected

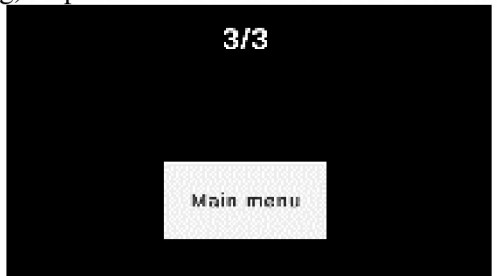

(h) Corrected fit after multiple iterations

(i) Score obtained through the iterative process

**Fig 1. The step-by-step interfaces of the PBL-VR training module.** Source: Authors' own work.

**Table 1. Feedback obtained through the pilot test.**

| General/ Module Design Issues | | |
|---|---|---|
| *Issue No.* | *Problem Identified* | *Applied Solution* |
| 1 | Useful only to students with existing pattern making and alteration knowledge (Expert 3). | The module starts with an explanation of how draglines and folds originate and how they can be corrected. |
| 2 | Initially, navigation of the training module was difficult (Expert 4). Red arrows pointing to pattern change were confusing (Experts 1 & 4). | A tutorial was added to the training module to explain how to choose the options. |
| 3 | Seeing the draglines and folds on white fabric was difficult (Experts 3 & 4). | The garments' fabric colors were changed. |
| Garment Specific Issues | | |
| *Garment* | *Problem Identified* | *Applied Solution* |
| **Pant** | The major fit issue at the front rise must be corrected before making other pattern modifications (Expert 4). | Excess at the front rise was eliminated. |
| **Sleeveless Top** | Error in explanation of fit correction in question 2 (Expert 1 & 5). | Replaced "Removing fabric from side seam" with "removing the fabric from armhole seam" (Expert 1). |
| **Skirt** | Waistband was not visible in the simulation (Expert 4). Body shape looks unrealistic (Expert 4). | Used different fabric colors for the waistband. Modified body shape to reduce hip circumference. |

based on their difficulty ranking into pre- and post-test [62]. Participants first completed the pre-test, followed by viewing the training module, and answering the post-test.

Interviews are often used as a method of training evaluation to gain an understanding of participants' perspectives on the training experience [63]. Following a previous research study in which participants were asked to use a VR application and then interviewed to get their feedback on the training tool, the participants were asked about their perceptions of the training module through a semi-structured interview [64]. Questions included "What did you think about the training module?", "Was the module helpful in increasing your fit evaluation and pattern correction skills?", "What challenges did you face during the training process? ", "What did you like and dislike the most about the training?", "How useful do you think the training module would be when training students?", "How can the training module be improved?", and "Do you have any other comments?".

## Data analysis

The data was analyzed through JMP Statistical Software. One-tailed paired t-tests were performed to determine whether ASVT and FCST post-test scores were significantly higher than the pre-test scores. To determine the impact of ASVT pre-test scores on the increase in FCST score, a linear regression model was created and the significance of the ASVT pre-test score was analyzed. The impact of gender and its interaction with pre-test scores were studied through another regression model. The main effect of gender and pre-test score was tested in a regression model followed by their interaction in the next model. The interviews were recorded and transcribed verbatim using Otter.ai. Inductive coding was performed using Microsoft Excel.

## Results

### Demographics

Forty participants were recruited for the study and divided into four age groups. Ten participants were in the age group younger than 20 years, 19 participants were in the age group 20–

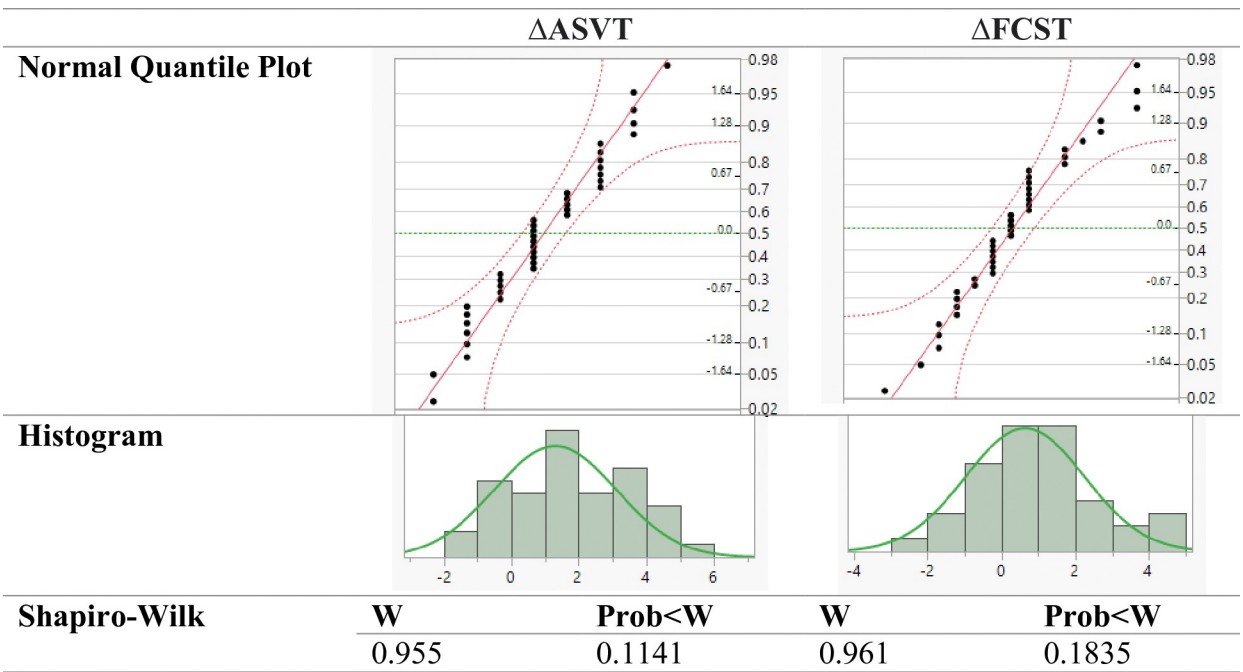

**Fig 2. Normality test for ASVT and FCST scores.**

25 years, eight participants were in the age group 25–30 years and three participants were above 30 years. Most of the participants were women (n = 25) followed by men (n = 15). 29 participants were Asian, nine participants were White, and two participants were Black.

## Testing assumptions

Each record was independent as participants worked individually throughout the process. Each participant's skill level was measured twice. Plots in Fig 2 indicated that the distribution of ΔASVT and ΔFCST scores was approximately bell-shaped and there were no extreme outliers. In addition, the Shapiro-Wilk test suggested that with a *p*-value bigger than .05, the data was normally distributed.

## Spatial visualization skill and fit correction skill learning (RQ1)

The mean difference between the pre-and post-test ASVT scores was 1.3 and the FCST score was 0.6. A strong correlation of 0.64 was observed between the ASVT pre-and post-test scores and a moderate correlation of 0.42 was observed between FCST pre-and post-test scores. One-sided paired t-test as seen in Fig 3 indicated a significant increase in apparel spatial visualization ($p < .0001$) as well as fit correction skill after the training ($p < .01$).

## Impact of prior spatial visualization skill on fit correction skill (RQ2)

As seen in Fig 4, the ASVT pre-test score, that is, the spatial visualization skill before undergoing training was a significant predictor of the Δ FCST score, that is, the increase in fit correction skill after undergoing the training ($p < 0.05$). A positive coefficient and the increasing slope of the line indicate that higher spatial visualization skills enable learners to realize higher improvement in terms of fit correction skills. A unit increase in ASVT pre-test score resulted in a 0.23 times higher increase in FCST score.

| | Post-test Mean | Pre-test Mean | Mean Diff | Std Error | Correlation |
|---|---|---|---|---|---|
| ASVT | 5.5 | 4.5 | 1.3 | .287 | .640 |
| FCST | 6.5 | 5.8 | .6 | .255 | .417 |
| | Upper 95% | Lower 95% | t | DF | Prob > t |
| ASVT | 1.880 | .720 | 4.531 | 39 | <.0001* |
| FCST | 1.153 | .122 | 2.499 | 39 | .0084* |

**Fig 3. Paired T-test for pre-and post-test scores.**

## Gender and learning outcomes (RQ3)

Regression models were developed for predicting the ASVT and FCST post-test scores using pre-test scores and gender. One model was developed with only the main effects and the second model was developed with the interaction term. As seen in Fig 5, in the model without interaction terms, the main effects for gender and pre-test score were found to be significant. Gender was found to be a significant predictor of the ASVT post-test score even when controlling for the ASVT pre-test score. As indicated by the regression plot, men scored lower than women. In the model with the interaction term, the interaction term was found to be statistically significant as well. When having low apparel spatial visualization skills, women saw a higher increase in skill as compared to men. With increasing skill, the difference between post-test scores reduced between participants of different genders. This might be a result of the ceiling effect where participants with high scores in the ASCT pre-test did not have much scope for improvement in the ASVT post-test. In conclusion, women benefitted more from the training module as compared to men. As seen in Fig 6, the regression models for predicting the FCST post-test score indicate that the gender or interaction of gender with the pre-test FCST score was not statistically significant. This implies that gender and its interaction with prior fit correction skill level did not impact the fit correction skill improvement because of the training.

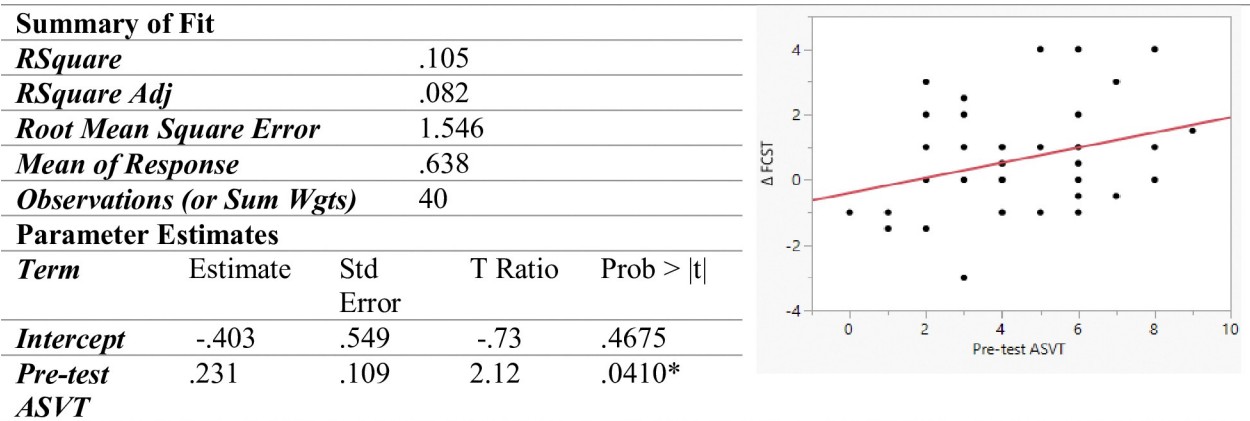

| Summary of Fit | | | | |
|---|---|---|---|---|
| *RSquare* | .105 | | | |
| *RSquare Adj* | .082 | | | |
| *Root Mean Square Error* | 1.546 | | | |
| *Mean of Response* | .638 | | | |
| *Observations (or Sum Wgts)* | 40 | | | |
| **Parameter Estimates** | | | | |
| *Term* | Estimate | Std Error | T Ratio | Prob > \|t\| |
| *Intercept* | -.403 | .549 | -.73 | .4675 |
| *Pre-test ASVT* | .231 | .109 | 2.12 | .0410* |

**Fig 4. Linear model predicting ΔFCST.**

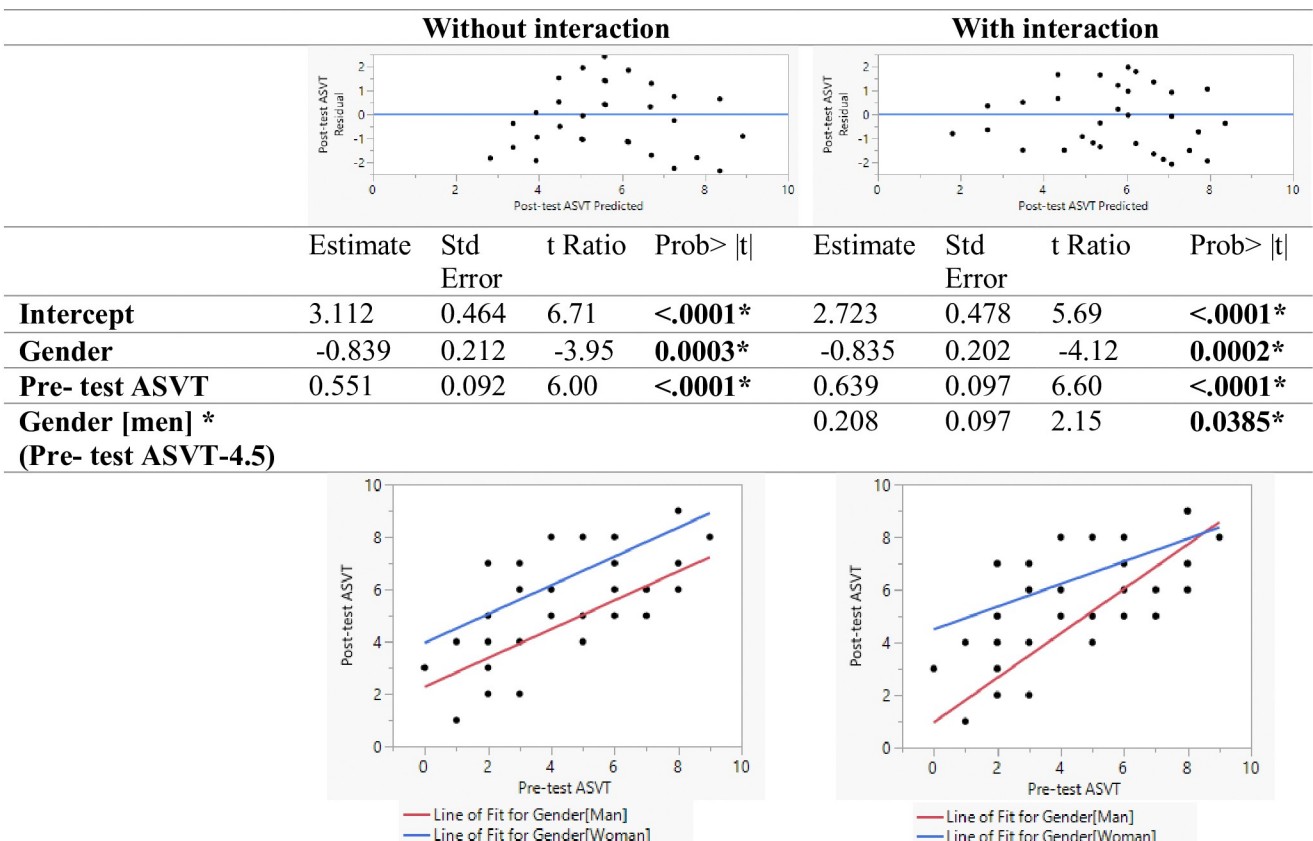

| | Without interaction | | | | With interaction | | | |
|---|---|---|---|---|---|---|---|---|
| | Estimate | Std Error | t Ratio | Prob> \|t\| | Estimate | Std Error | t Ratio | Prob> \|t\| |
| **Intercept** | 3.112 | 0.464 | 6.71 | **<.0001*** | 2.723 | 0.478 | 5.69 | **<.0001*** |
| **Gender** | -0.839 | 0.212 | -3.95 | **0.0003*** | -0.835 | 0.202 | -4.12 | **0.0002*** |
| **Pre- test ASVT** | 0.551 | 0.092 | 6.00 | **<.0001*** | 0.639 | 0.097 | 6.60 | **<.0001*** |
| **Gender [men] \* (Pre- test ASVT-4.5)** | | | | | 0.208 | 0.097 | 2.15 | **0.0385*** |

**Fig 5. Linear model predicting ASVT score.**

## Participant perceptions of the PBL-VR training (RQ4)

Based on the coded responses, seven themes were identified. The first theme, "Level of Effectiveness", revealed that the participants with prior pattern making experience found the PBL-VR training a useful refresher. Participants without pattern making experience directly applied the information in the later stage of training. Participants appreciated the "step-by-step approach" (Participant #1) of correcting garments and the "good amount of examples" (Participant #18) especially because they resembled real-world garment fitting sessions. Participant 5 emphasized the effectiveness of the problem-based learning approach as follows:

> "When we are working on projects [after completing the training] we will be able to better understand the difficulties we're having in our project and how it can be improved. And just having that visual knowledge and insight into what cues we can look for in the fit of a garment, I think is really helpful."

It was a unanimous opinion among participants with varying levels of experience that the training module could be used as a supplement but not a replacement for traditional in-person instruction. Participants believed that the training module would be "helpful to students combined with physical training" (Participant #30). Additionally, participants mentioned that the module was useful if the user already has a foundational level of understanding (Participants #20 and #34).

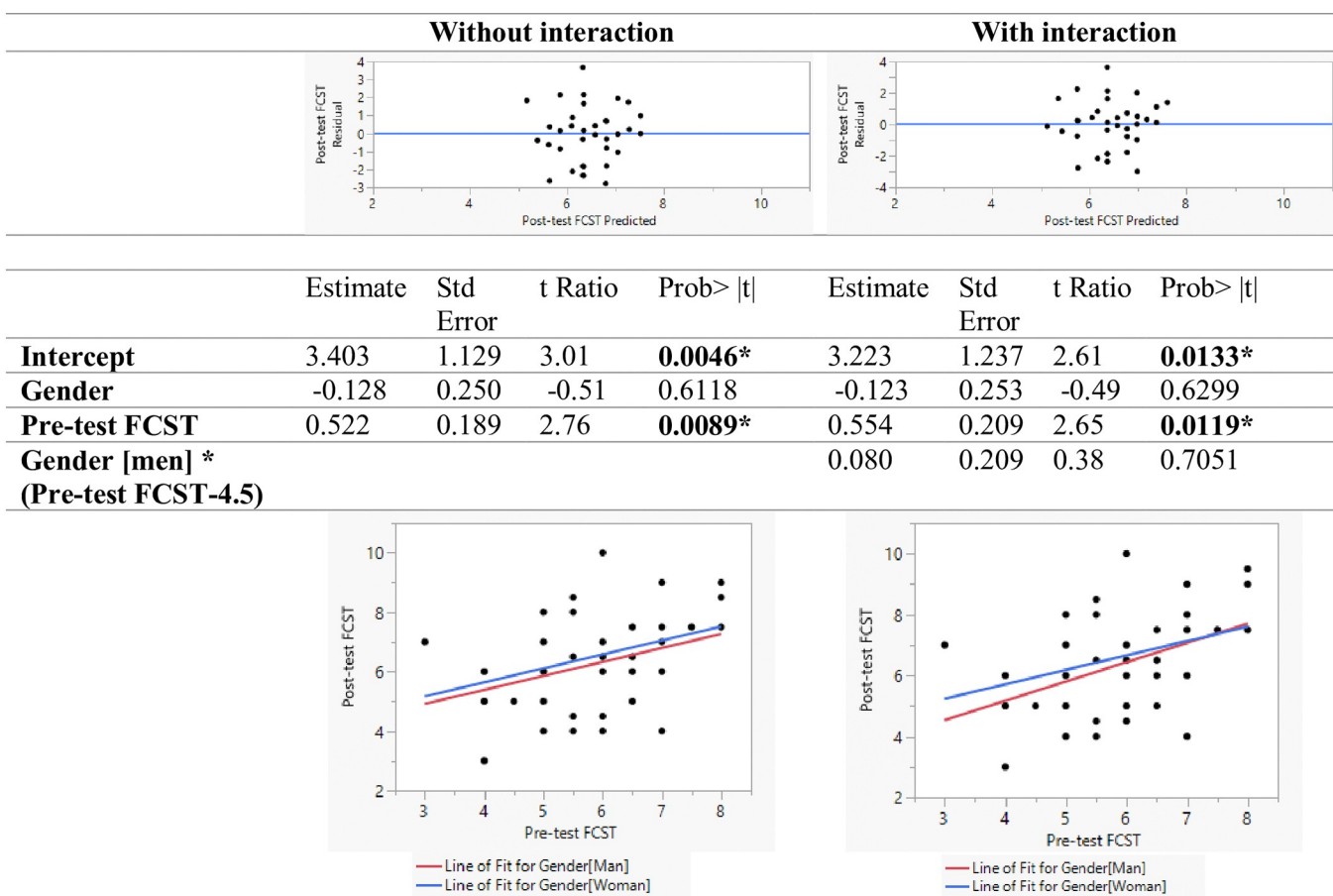

| | Without interaction | | | | With interaction | | | |
|---|---|---|---|---|---|---|---|---|
| | Estimate | Std Error | t Ratio | Prob> \|t\| | Estimate | Std Error | t Ratio | Prob> \|t\| |
| **Intercept** | 3.403 | 1.129 | 3.01 | **0.0046*** | 3.223 | 1.237 | 2.61 | **0.0133*** |
| **Gender** | -0.128 | 0.250 | -0.51 | 0.6118 | -0.123 | 0.253 | -0.49 | 0.6299 |
| **Pre-test FCST** | 0.522 | 0.189 | 2.76 | **0.0089*** | 0.554 | 0.209 | 2.65 | **0.0119*** |
| **Gender [men] * (Pre-test FCST-4.5)** | | | | | 0.080 | 0.209 | 0.38 | 0.7051 |

**Fig 6. Linear model predicting the FCST score.**

The second theme emerged as "Visualization Speed". Participants who had taken pattern making classes before appreciated the ability to immediately see the impact of a pattern change as compared to the in-studio process where changes take a considerable amount of time and material. They called the training "more sustainable" (Participant #31) and "time-saving" (Participant #9). For instance, one of the participants mentioned:

> "I liked how you could visualize immediately what the fit changes would do because I learned it in an in-person space, and you couldn't see those changes fast. And this [training module] just sped up the process and made it a lot easier to register the information." (Participant #21)

The third theme was "Variation in body shapes and garments". In pattern making classes students usually work on a dress form with an hourglass figure and do not have the opportunity to test garment fitting on different body shapes. The fitting process is unique for each body shape based on the relationship between the clothing and the body. Several participants appreciated the variety in terms of clothing and body shapes of the models. Participants described it as "very helpful" (Participant #9). One of the participants mentioned:

"It was nice that there were different bodies too, not just like, one mannequin, which is what we use, like a size eight. Yes, people's bodies aren't shaped the same." (Participant #37)

The fourth theme emerged as "Showing Weak Points." By going through the interactive PBL process, participants could recognize the areas that they needed to work on. For instance, some participants identified their area of weakness as: "difficult to tell if the garment was too small or too big" (Participant #31), "couldn't tell if it was a dragline or fold" (Participant #20), "hard to tell if the shoulder should be changed or the armhole" (Participant #35), "confusing if one side of the shoulder should be lifted or both" (Participant #37) and so on. The multiple-choice questions allowed participants to identify the areas where they were going wrong. One of the participants said:

"I liked how it [training module] would give you multiple guesses because a lot of times, I would think that one thing is right. So it was good to have multiple chances to figure out the mistakes I kept making." (Participant #35)

The fifth theme was the "Transfer of Skills to the Real World". One of the participants with no garment fitting experience described the module as "eye-opening" (Participant #21) and several of them mentioned that they started looking at their garments and were able to pinpoint the fit issues. For instance, one of the participants mentioned:

"Why does it like look this right here on my shirt? sagging? Oh, clearly, that means probably a bit too big in sleeves." (Participant #33)

The sixth theme was "Potential Uses". The potential use case as described by participants was as a "self-teaching" (Participant #33) tool for students accompanied by an in-person session. Participants talked about the usability of the module as a distance learning tool, in the industry as well as to educate customers about garment fit. They described the module as a "helpful standalone educational tool for online based learning" (Participant #27), "useful for non-technical design professionals who could benefit from additional pattern making experience" (Participant #27), "tool for designers who don't have as much pattern making experience", (Participant #27) "useful assessment tool without making people do much" (Participant #11) or "for educating customers" (Participant #6).

The seventh and last theme was "Suggestions for Improvement". The training module only showed the change in garment simulation when the user selected the right pattern modification option. Several participants expressed the importance of enhanced interactivity similar to a "dress up game" (Participant #36), for example, seeing the impact of selecting a wrong pattern modification option to improve learning. Another participant (Participant #2) mentioned it might be helpful to build levels such as beginner, intermediate, and advanced to cater to people with varying levels of experience.

## Discussions

With rapid prototyping and reduced lead times in the fashion industry, designers must be prepared to efficiently use digital garment simulations. Through the current study, a novel training module to improve the fit correction skills of designers by leveraging the latest technology was developed. The training modules were aimed at students who would like to improve their knowledge of garment fit regardless of their prior experience. The findings of the study indicated that a short 20-minute PBL session in VR can improve the apparel spatial visualization as well as fit correction skills of learners. The tasks closely resemble the real-world fit sessions and provide the experience of fitting garments on different body shapes as well as different

garments. They also enable seeing the immediate impact of pattern changes on the 3D garment fit. The presence of virtual garments provides a visual aid in developing an intuition about the pattern change that will correct the fit of the garment. This supports previous research that indicates apparel design training improves apparel spatial visualization skills of the learners [53] as well as studies that suggest problem-based education improves learning outcomes [65].

Prior apparel spatial visualization skills were found to play a crucial role in the learning of garment fit correction skills. Participants who had higher apparel spatial skills before training could probably better visualize and connect the 2D patterns with 3D garments. The findings indicated that apparel spatial visualization skill is a prerequisite to learning efficiently from digital garment simulations. Therefore, participants should first be exposed to physical garment development and fitting sessions before being introduced to the digital problem-based learning module. While this finding agrees with the interview feedback, which suggests that prior experience with physical garments is required to learn effectively from PBL-VR modules, it is contrary to previous research that indicates learners with low spatial abilities are more positively affected [66]. This difference might be attributed to the difference in the field of education. Complex visualization and spatial skills are required in apparel design education.

Gender was found to have a significant impact on apparel spatial visualization skills but not the fit correction skill. On the lower end of spatial visualization ability, women saw a higher increase in spatial visualization skills because of the training. At higher levels of spatial visualization ability, the difference in scores between men and women disappeared. This might be because the training module included women's garments which made it easier for women to visualize and grasp concepts. Apart from this, gender matching of the avatar with the participant could have played a role in increasing the performance of women [67].

Overall, participants appreciated the training structure, immediate feedback, visual aid, and variation in body shapes and garment styles. They acknowledged the importance of being able to see immediate changes and their advantages in terms of saving material and time. Additionally, participants found the modules to be a good replication of real fitting sessions where several steps are required to correct the fit. They believed that the training was good practice as well as a supplement for physical in-person classes but not a replacement. This is in accordance with previous studies that indicated VR would rather supplement than replace traditional training [68].

## Conclusions

In the current study, a novel problem-based learning module for apparel fit correction was developed and tested with a pre and post-test approach. Expert feedback helped improve the content as well as the usability of the training module. The PBL questions were designed based on relevance to real-world garment design challenges and the need to develop spatial visualization and fit correction skills required for using 3D CAD software programs. The results of the study proved the usefulness of digital problem-based learning with apparel garment simulations and a problem-based approach. The findings indicated a significant improvement in fit correction skills and spatial visualization skills post-training. Students with higher spatial visualization skills pre-training saw significantly better improvement in fit correction skills implying that prior apparel spatial knowledge is crucial to gain benefits from digital learning modules. Gender as well as the interaction between gender and apparel spatial skills was significant when predicting the performance on the spatial visualization skill test. When having lower apparel spatial abilities, women saw a significantly higher improvement than men. The difference in performance between genders was reduced when participants had higher spatial abilities. Several participants who had completed traditional pattern making courses

appreciated the benefit of using problem-based learning as a method of evaluating garment fit. The VR-based PBL approach proved effective because it ensured engagement, provided immediate feedback, and enabled quick iteration. One of the largest benefits of the virtual training as pointed out by participants was that it enabled seeing the impact of pattern changes on the 3D garment fit within seconds which would possibly take several minutes if not hours when working with actual garments. In addition, they appreciated the iterative process which was a more realistic approach to fit correction than making all the pattern changes in one step.

The findings of this study must be generalized with caution because of the small sample size, which may undermine the statistical power, and lack of diversity in participant demographics. It should be noted that all participants were college students. Even though the study did not mitigate potential bias and subjectivity in recording the participants' perceptions, we took several measures to diminish the impact. Likert-type scale questionnaires that use a consistent framework for participants to share their responses were used. Moreover, validity checks such as reverse-coded items in the questionnaire to identify biased responses were introduced. We further triangulated the data by comparing the responses to questionnaires as well as interviews to reduce the inherent bias of any particular method. The interviews provided an opportunity to ask participants to elaborate on their responses and gain a comprehensive understanding of their perceptions. In the current study, the performance of users was tested right after the training, that is, the short-term learning effects of the training were analyzed. The post-test was given to participants immediately after the training which could have led to mental exhaustion and fatigue. Providing a day or two of rest might provide a better understanding of the impact of training. Additionally, a delayed retention test could provide useful insight into the long-term effects of the training. Future studies could improve the feedback mechanism in the VR training module by introducing features such as interactive coaching elements or peer collaboration. In the current study, we used a within-subjects design to test the impact of training. Each group's outcome was tested pre-and post-training. Further, the study aimed to evaluate whether VR training can provide an improvement from the current method of training which was simulated using the 2D video scenario. Therefore, the focus was to test the effect of the intervention. A future study would benefit from having a control group. Despite significant advancements in VR technology, there are still several challenges, such as simulator sickness that can hinder immersive experience and potentially impact learning outcomes [57, 69]. Therefore, simulator sickness or psychological discomfort should be examined in future studies. While the current study mainly focused on learning outcomes associated with VR technology, it is essential to acknowledge the cost implications [70] inherent in its adoption. Even with desktop VR, the software development costs may pose a challenge. Additionally, potential barriers to access and adoption of VR technology in educational institutions, such as infrastructure requirements should be considered in future studies. There are several challenges associated with incorporating VR technology into educational settings [70]. Requirements such as equipment availability, technical support, careful planning, and integration within existing curriculum frameworks pose barriers to widespread adoption.

It is essential to note that the participants' perceptions may be subject to the novelty effect. The unfamiliarity with 3D interactive models shown in the training module might lead to excitement that could impact the experience as well as attitude. However, even participants who had years of experience with apparel 3D CAD programs appreciated the training module and pointed out several benefits. While the current study focused on identifying the potential benefits of using a prototype VR training module, it is important to acknowledge the inherent technical limitations. One such limitation was the inability to fully interact with the garment itself. Due to the nature of the prototype and constraints within the Unity platform, the participants could not manipulate the fabric of the garment like they potentially could in apparel

digital prototyping software. Even though our findings primarily apply to the academic setting, we recognize the importance of future research to explore the applicability of these findings in an industry setting.

While differences in learning styles and preferences may affect the benefit gained from the training, the current study aimed to provide a general assessment across a diverse group of participants. Future studies should explore the possibility of customizing VR experiences to accommodate different learning styles and explore potential differences in learning outcomes based on participants' prior exposure to VR technology. Although we assessed participants' prior apparel spatial visualization skill to explore potential differences in learning outcomes based on prior familiarity with apparel 2D to 3D pattern transformations, it is important to acknowledge that this does not capture the potential difference introduced because of familiarity with the VR training platform.

The focus of the current study was to evaluate the immediate impact of the training module on performance. Future studies should examine the long-term impacts of the training module which is crucial to gain an understanding of the sustainability and scalability of implementing the VR training module into apparel education. Moreover, as indicated by a previous study, the accessibility barriers in VR should be examined in more detail [71]. Therefore, further research studies in designing inclusive user interfaces and providing alternate modes of interaction would be needed to improve the training modules. Additionally, future research should identify ways to protect personal information and ensure secure data storage within the VR training module. It is crucial to consider the potential drawbacks associated with overreliance on VR simulations such as reduced hands-on experience and practical skill development. Understanding and accommodating cultural and contextual differences are essential for designing VR training programs that are both acceptable and effective across diverse populations. Future studies should consider factors such as cultural attitudes towards technology [72], and language [73], to improve the developed VR training modules that can resonate with learners from various cultural backgrounds.

The purpose of the current study was to serve as a preliminary investigation into the potential of using VR as a training tool and assess feasibility, acceptability, and learning gains before conducting longer-term studies. Additionally, we focused on the short-term effect of VR rather than long-term skill improvement. This provided insights into the initial learning curve as well as user acceptance. Short-term training sessions can be a part of a larger training program. While a single short-term session cannot holistically capture learning gains, it can serve as a starting point for longer interventions. The novel modules are expected to substitute the years of real-world patternmaking experience required to achieve fit correction skills for various body shapes and sizes. As the next step, the effect of the medium of instruction will be analyzed as well as other constructs such as motivation and cognitive benefit impacting learning through virtual reality will be explored. In addition, the transfer of skills to the real-world task will be analyzed.

## Supporting information

**S1 Fig. Iterative fit correction tasks.**
(PDF)

## Acknowledgments

We would like to thank Mr. Kevin Kim for his help with developing the training modules in virtual reality and Mr. Matt Thomas from Cornell University Statistical Unit for their guidance on data analysis.

## Author Contributions

**Conceptualization:** Aditi Galada, Fatma Baytar.

**Data curation:** Aditi Galada.

**Formal analysis:** Aditi Galada.

**Funding acquisition:** Aditi Galada, Fatma Baytar.

**Methodology:** Aditi Galada, Fatma Baytar.

**Project administration:** Fatma Baytar.

**Resources:** Fatma Baytar.

**Supervision:** Fatma Baytar.

**Validation:** Aditi Galada.

**Writing – original draft:** Aditi Galada.

**Writing – review & editing:** Aditi Galada, Fatma Baytar.

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
