## [Decision Letter · Decision Letter 0]

13 Feb 2024

PONE-D-23-25449Design and evaluation of a problem-based learning VR module for apparel fit correction trainingPLOS ONE

Dr. Baytar,

Thank you for submitting your manuscript to PLOS ONE. After careful consideration, we feel that it has merit but does not fully meet PLOS ONE’s publication criteria as it currently stands. Therefore, we invite you to submit a revised version of the manuscript that addresses the points raised during the review process.

We look forward to receiving your revised manuscript.

Kind regards,

Saliha Karadayi-Usta, PhD

Academic Editor

PLOS ONE

Journal Requirements:

Reviewers' comments:

Reviewer's Responses to Questions

**Comments to the Author**

1. Is the manuscript technically sound, and do the data support the conclusions?

Reviewer #1: Yes

Reviewer #2: Yes

2. Has the statistical analysis been performed appropriately and rigorously? 

Reviewer #1: Yes

Reviewer #2: Yes

3. Have the authors made all data underlying the findings in their manuscript fully available?

Reviewer #1: Yes

Reviewer #2: No

4. Is the manuscript presented in an intelligible fashion and written in standard English?

Reviewer #1: Yes

Reviewer #2: Yes

5. Review Comments to the Author

Reviewer #1: This research demonstrates the potential of problem-based learning and digital simulations in addressing challenges related to apparel fit correction. While the study provides valuable insights, further research with a larger and more diverse participant pool, as well as long-term assessments, will strengthen the validity of the findings. The study's focus on real-world applicability and future research directions are particularly commendable, as they have the potential to reshape education and practice in the field of apparel design.

for additional considerations, this research still need an expert feedback during the development process of the learning module to ensuring its quality and relevance especially on the design skills and also to see how well these virtual skills transfer to actual garment production. This iterative process demonstrates a commitment to refining the training content. and also for the validation in this research, these results due to the small sample size and the participants' specific demographic so its needs more diverse samples would help validate these findings.

Reviewer #2: This article emphasizes the importance of using problem-based learning and virtual reality in clothing design training. The research aims to evaluate the effectiveness of a virtual reality-based training module to improve clothing design students' fit skills. This study demonstrates how virtual reality technology can improve education in specialized fields such as clothing design. The authors also note that more research is needed to investigate how virtual reality learning environments influence other factors such as motivation and cognitive benefits. This study also suggests that future research could investigate how skills learned in VR can be transferred to real-world applications.

Some aspect to be reviewed:

1. Lack of diversity in participant demographics may limit the generalizability of the findings.

2. Small sample size undermines the statistical power and reliability of the results.

3. The quasi-experimental design may introduce confounding variables, affecting the validity of the conclusions.

4. Limited longitudinal data collection prevents the assessment of long-term retention and application of skills.

5. The reliance on self-reported measures for participant perceptions introduces potential bias and subjectivity.

6. Potential selection bias due to voluntary participation may skew the results towards individuals more inclined towards technology.

7. The study does not address potential ethical concerns related to VR technology, such as cybersickness or psychological discomfort.

8. Lack of control group prevents comparison with alternative training methods, limiting the ability to evaluate the relative effectiveness of the VR module.

9. The study does not consider variations in VR hardware and software, which may impact user experience and training outcomes.

10. The duration of the VR training session is not sufficient to adequately assess skill improvement.

11. The absence of follow-up assessments hinders understanding of skill retention over time.

12. The study overlooks potential cost implications associated with implementing VR technology in educational settings.

13. Limited information is provided regarding the development process and validation of the VR training module.

14. The study does not explore potential adverse effects of VR exposure, such as disorientation or motion sickness.

15. Participant perceptions may be influenced by novelty effects.

16. The study does not address potential technical issues or limitations of the VR platform.

17. The focus on college students may neglect the perspectives and needs of professionals or individuals outside academic settings.

18. The training module's effectiveness may vary based on individual learning styles and preferences, which are not accounted for in the study.

19. The study does not consider potential cultural or contextual differences that may influence the acceptability and effectiveness of VR training.

20. Lack of information on the reliability and validity of the assessment tools used to measure spatial visualization skills and fit correction abilities.

21. The study does not explore potential biases introduced by the researchers or instructors involved in the development and evaluation of the VR module.

22. The absence of objective performance metrics limits the ability to quantify skill improvement objectively.

23. The study does not investigate potential long-term implications of relying on VR technology for education and training.

24. Limited information is provided on the accessibility and inclusivity of the VR training module for individuals with disabilities.

25. The study does not address potential concerns related to data privacy and security in VR learning environments.

26. The absence of detailed feedback mechanisms within the VR module may hinder participants' ability to receive personalized guidance and support.

27. The study does not explore potential differences in learning outcomes based on participants' prior exposure to VR technology.

28. The study does not consider potential barriers to access and adoption of VR technology in educational institutions, such as infrastructure requirements or cost constraints.

29. The study does not investigate potential negative consequences of overreliance on VR simulations, such as reduced hands-on experience or practical skills development.

30. The study lacks discussion on potential limitations and challenges in implementing VR-based training modules within existing curriculum frameworks.

31. References are wrong, they should be revised and updated based according to the recent developments.

32. Figures should be enhanced. They should be properly cited in text.

33. Abbreviations should be used appropriately.

6. PLOS authors have the option to publish the peer review history of their article (what does this mean?). If published, this will include your full peer review and any attached files.

Reviewer #1: No

Reviewer #2: **Yes: **Peren Jerfi Canatalay

---

## [Author Response · Author response to Decision Letter 0]

4 Apr 2024

We updated the revised manuscript's formatting accordingly.

Thank you, we deleted this information from the manuscript. 

All data, code, and log files are available from the Cornell eCommons data archive. The files can be accessed by following this DOI link: Data from: Design and evaluation of a problem-based learning VR module for apparel fit correction training. [dataset] Cornell University eCommons Repository. https://doi.org/10.7298/8wsv-0q41

Thank you, we used the correct title this time. 

Our ethics statement now only appears in the Methods section in the revised version.

We corrected the caption for our Supporting Information files, which were submitted as a separate file.

Reviewers' comments:

Reviewer's Responses to Questions

Comments to the Author

1. Is the manuscript technically sound, and do the data support the conclusions?

Reviewer #1: Yes

Reviewer #2: Yes

2. Has the statistical analysis been performed appropriately and rigorously? 

Reviewer #1: Yes

Reviewer #2: Yes

3. Have the authors made all data underlying the findings in their manuscript fully available?

Reviewer #1: Yes

Reviewer #2: No

We deposited our data to eCommons in this revised version. 

4. Is the manuscript presented in an intelligible fashion and written in standard English?

Reviewer #1: Yes

Reviewer #2: Yes

5. Review Comments to the Author

Reviewer #1: This research demonstrates the potential of problem-based learning and digital simulations in addressing challenges related to apparel fit correction. While the study provides valuable insights, further research with a larger and more diverse participant pool, as well as long-term assessments, will strengthen the validity of the findings. The study's focus on real-world applicability and future research directions are particularly commendable, as they have the potential to reshape education and practice in the field of apparel design.

for additional considerations, this research still need an expert feedback during the development process of the learning module to ensuring its quality and relevance especially on the design skills and also to see how well these virtual skills transfer to actual garment production. This iterative process demonstrates a commitment to refining the training content. and also for the validation in this research, these results due to the small sample size and the participants' specific demographic so its needs more diverse samples would help validate these findings.

Reviewer #2: This article emphasizes the importance of using problem-based learning and virtual reality in clothing design training. The research aims to evaluate the effectiveness of a virtual reality-based training module to improve clothing design students' fit skills. This study demonstrates how virtual reality technology can improve education in specialized fields such as clothing design. The authors also note that more research is needed to investigate how virtual reality learning environments influence other factors such as motivation and cognitive benefits. This study also suggests that future research could investigate how skills learned in VR can be transferred to real-world applications.

Some aspect to be reviewed:

1. Lack of diversity in participant demographics may limit the generalizability of the findings.

Thank you for pointing that out. We updated the following line in our manuscript to reflect this limitation: “The findings of this study must be generalized with caution because of the small sample size, which may undermine the statistical power, and lack of diversity in participant demographics. It should be noted that all participants were college students.” 

2. Small sample size undermines the statistical power and reliability of the results.

We were able to recruit 40 participants for our study. This limitation was addressed under the limitations section as explained in Comment #1.

3. The quasi-experimental design may introduce confounding variables, affecting the validity of the conclusions.

We understand that the quasi-experimental design reduces the internal validity, but we chose this approach because collecting measurements before and after training enabled us to create strong inferences. Our focus in the current study was to compare the change in performance within the groups over time. It enabled us to create more representative and diverse samples within a limited time frame and cost.

4. Limited longitudinal data collection prevents the assessment of long-term retention and application of skills.

In the Conclusion section, we had already acknowledged that the current study focused on the short-term effects of training and recommended another study to assess the long-term impact as follows: “Additionally, a delayed retention test could provide useful insight into the long-term effects of the training.”

5. The reliance on self-reported measures for participant perceptions introduces potential bias and subjectivity.

We addressed your comment in the text as follows: “Even though the study did not mitigate potential bias and subjectivity in recording the participants’ perceptions, we took several measures to diminish the impact. Likert-type scale questionnaires that use a consistent framework for participants to share their responses were used. Moreover, validity checks such as reverse-coded items in the questionnaire to identify biased responses were introduced. We further triangulated the data by comparing the responses to questionnaires as well as interviews to reduce the inherent bias of any particular method. The interviews provided an opportunity to ask participants to elaborate on their responses and gain a comprehensive understanding of their perceptions.”

6. Potential selection bias due to voluntary participation may skew the results towards individuals more inclined towards technology.

The concern of potential selection bias due to voluntary participation is valid but was addressed through providing incentives such as monetary compensation as well as credit in courses for participation that appealed to a wide range of individuals and not just people interested in technology. We added our response under Data Collection section: “The focus on college students was intended to provide a targeted examination of the potential benefits of incorporating VR into the curriculum. Providing incentives such as monetary compensation as well as credit in courses for participation that appealed to a wide range of individuals and not just people interested in technology helped avoid selection bias.”

7. The study does not address potential ethical concerns related to VR technology, such as cybersickness or psychological discomfort.

The focus of the current study was on a specific aspect of VR technology, which was its effectiveness in training. This study is expected to serve as a foundation for the use of VR in the apparel digital prototyping field. We believe that addressing ethical concerns should be a topic for future research building on the insights gained by the current study. We addressed this comment in Conclusions as follows: “Despite significant advancements in VR technology, there are still several challenges, such as simulator sickness that can hinder immersive experience and potentially impact learning outcomes [69]. Therefore, simulator sickness or psychological discomfort should be examined in future studies.”

8. Lack of control group prevents comparison with alternative training methods, limiting the ability to evaluate the relative effectiveness of the VR module.

In the current study, we used a within-subjects design to test the impact of training. Each group’s outcome was tested pre-and post-training. Further, the study aimed to evaluate whether VR training can provide an improvement from the current method of training which was simulated using the 2D video scenario. Therefore, the focus was to test the effect of the intervention. A future study would benefit from having a control group. We added this in the recommendation for future studies paragraph in Conclusion. 

9. The study does not consider variations in VR hardware and software, which may impact user experience and training outcomes.

The current study focused on assessing the effectiveness of VR as a medium of instruction in the specific context of training in apparel fit correction skills. Given the wide range of VR technologies available, it was not in the scope of the current study to account for the hardware and software variations. 

10. The duration of the VR training session is not sufficient to adequately assess skill improvement.

The purpose of the current study was to serve as a preliminary investigation into the potential of using VR as a training tool and assess feasibility, acceptability, and learning gains before conducting longer-term studies. Additionally, we focused on the short-term effect of VR rather than long-term skill improvement. This provided insights into the initial learning curve as well as user acceptance. Short-term training sessions can be a part of a larger training program. While a single short-term session cannot holistically capture learning gains, it can serve as a starting point for longer interventions. Our response was added under Conclusions. 

11. The absence of follow-up assessments hinders understanding of skill retention over time.

The focus of our study was to determine short-term learning gains rather than assessing skill retention over a period. As explained in Comment #10, the effectiveness of the training in the short term provided us insights into the initial learning curve as well as user acceptance. While a single short-term session cannot holistically capture learning gains, it can serve as a starting point for longer interventions. Our response was added as a suggestion for future studies under Conclusions.

12. The study overlooks potential cost implications associated with implementing VR technology in educational settings.

It is crucial to acknowledge that VR technology can be expensive to implement. In the current study we focused on desktop virtual reality which does require a virtual reality headset and can instead be viewed through a monitor but there are still costs associated with software development that need to be addressed. We added the following lines in the manuscript to recognize the cost implications: “While the current study mainly focused on learning outcomes associated with VR technology, it is essential to acknowledge the cost implications [70] inherent in its adoption. Even with desktop VR, the software development costs may pose a challenge.”

13. Limited information is provided regarding the development process and validation of the VR training module.

We provided detailed information under the Methodology section as follows. Additional information on developing the test instrument and validating it was submitted to another journal and accepted with minor revisions. It has not been published yet, therefore, we cited the thesis of the first author in this paper (details were omitted from the blind review). 

“To develop training modules, previous literature on indicators of a good fit, recognizing fit issues and rectifying them, previous training modules developed to educate apparel design students, and the tools used to measure domain-specific learning were analyzed thoroughly. The training modules were designed to teach the participants to use the ‘recognition-primed decision model’ [47] when determining what pattern modification option would correct the garment fit. By following the Experiential Learning Theory, the modules were designed to provide concrete experiences on fit correction, allow learners to re

---

## [Decision Letter · Decision Letter 1]

9 Jul 2024

PONE-D-23-25449R1Design and evaluation of a problem-based learning VR module for apparel fit correction trainingPLOS ONE

Dear Dr. Baytar,

Thank you for submitting your manuscript to PLOS ONE. After careful consideration, we feel that it has merit but does not fully meet PLOS ONE’s publication criteria as it currently stands. Therefore, we invite you to submit a revised version of the manuscript that addresses the points raised during the review process. Please attend to the minor suggestions made by reviewers #5 and #8.

We look forward to receiving your revised manuscript.

Kind regards,

Avanti Dey, PhD

Staff Editor

PLOS ONE

Journal Requirements:

Additional Editor Comments (if provided):

Reviewers' comments:

Reviewer's Responses to Questions

**Comments to the Author**

1. If the authors have adequately addressed your comments raised in a previous round of review and you feel that this manuscript is now acceptable for publication, you may indicate that here to bypass the “Comments to the Author” section, enter your conflict of interest statement in the “Confidential to Editor” section, and submit your "Accept" recommendation.

Reviewer #3: All comments have been addressed

Reviewer #4: All comments have been addressed

Reviewer #5: (No Response)

Reviewer #6: (No Response)

Reviewer #7: (No Response)

Reviewer #8: All comments have been addressed

2. Is the manuscript technically sound, and do the data support the conclusions?

Reviewer #3: Partly

Reviewer #4: Yes

Reviewer #5: Yes

Reviewer #6: Yes

Reviewer #7: Yes

Reviewer #8: Yes

3. Has the statistical analysis been performed appropriately and rigorously? 

Reviewer #3: Yes

Reviewer #4: Yes

Reviewer #5: Yes

Reviewer #6: Yes

Reviewer #7: Yes

Reviewer #8: Yes

4. Have the authors made all data underlying the findings in their manuscript fully available?

Reviewer #3: Yes

Reviewer #4: No

Reviewer #5: Yes

Reviewer #6: Yes

Reviewer #7: Yes

Reviewer #8: Yes

5. Is the manuscript presented in an intelligible fashion and written in standard English?

Reviewer #3: Yes

Reviewer #4: Yes

Reviewer #5: Yes

Reviewer #6: Yes

Reviewer #7: Yes

Reviewer #8: Yes

6. Review Comments to the Author

Reviewer #3: An updated manuscript addressing previous comments and suggestions was evaluated positively. The updated submission demonstrates significant improvement and provides valuable insights relevant to the research community.

Reviewer #4: The authors have made all revisions in the new version. This manuscript is well-written and easy to read.

Reviewer #5: This paper proposed to develop a very interesting problem-based learning model for apparel fit correction using VR technology. The presented research certainly facilitates the promotion of learning outcomes of garment pattern design course. However, after reading the paper, one point still needs the author’s attention: over 70% of references are from five years ago. It is suspected that the authors are really aware of all the most recent literatures related to this topic.

Reviewer #6: This is an interesting paper that illustrates approaches that can be developed to better understand fit and fit issues.

Why is "apparel fit" not among the papers keywords?

Reviewer #7: The article deals with the design and evaluation of a problem-based VR learning module for garment fit correction training. The article is well conceptualised and gives an insight into the nature of problem-based learning and shows potential in the field of digital fashion design.

The article has been thoroughly corrected and amended. I have no additional comments.

Reviewer #8: By utilizing virtual reality technology, this paper proposed an interactive problem-based learning approach to help the students in understanding garment design. They ask the student to apply 3D garment design software such as CLO3D to realize 3D garment fit in a problem-based learning manner. Experimental results show that such approach can improve the spatial visualization and fit correction skills. In the revised version, the authors responded well to the previous round of review comments.

If the author can explain why these problem issues were designed and why they are effective, it can make the conclusions of the paper more feasible.

7. PLOS authors have the option to publish the peer review history of their article (what does this mean?). If published, this will include your full peer review and any attached files.

Reviewer #3: No

Reviewer #4: **Yes: **Omid Abbaszadeh

Reviewer #5: No

Reviewer #6: No

Reviewer #7: No

Reviewer #8: No

---

## [Author Response · Author response to Decision Letter 1]

24 Jul 2024

Dear Editor,

Thank you for the opportunity to revise our manuscript and for the reviewers' comments. Please see our responses to the minor revision suggestions from reviewers #$4, #5, #6 and #8 below:

4. Have the authors made all data underlying the findings in their manuscript fully available?

Reviewer #3: Yes

Reviewer #4: No 

All data, code, and log files are available from the Cornell eCommons data archive. The files can be accessed by following this DOI link: Data from: Design and evaluation of a problem-based learning VR module for apparel fit correction training. [dataset] Cornell University eCommons Repository. https://doi.org/10.7298/8wsv-0q41

Reviewer #5: Yes

Reviewer #6: Yes

Reviewer #7: Yes

Reviewer #8: Yes

6. Review Comments to the Author

Reviewer #5: This paper proposed to develop a very interesting problem-based learning model for apparel fit correction using VR technology. The presented research certainly facilitates the promotion of learning outcomes of garment pattern design course. However, after reading the paper, one point still needs the author’s attention: over 70% of references are from five years ago. It is suspected that the authors are really aware of all the most recent literatures related to this topic.

Thank you. We went through the manuscript and updated 13 relevant references with the most recent literature. We highlighted them in the references. We kept the rest the way they were because they justified our methods and informed our discussions and implications of our findings. 

Reviewer #6: This is an interesting paper that illustrates approaches that can be developed to better understand fit and fit issues.

Why is "apparel fit" not among the papers keywords?

Thank you, we added “Apparel fit” to the keywords.

Reviewer #8: By utilizing virtual reality technology, this paper proposed an interactive problem-based learning approach to help the students in understanding garment design. They ask the student to apply 3D garment design software such as CLO3D to realize 3D garment fit in a problem-based learning manner. Experimental results show that such approach can improve the spatial visualization and fit correction skills. In the revised version, the authors responded well to the previous round of review comments.

If the author can explain why these problem issues were designed and why they are effective, it can make the conclusions of the paper more feasible.

Thank you. In the Literature review, between lines 51-61, we introduced our rationale for why Problem-Based Learning is important and effective as follows: “VR provides a fun, interesting, and immersive way of learning [13]. Integration of garment simulations in a PBL module can allow higher interactivity as well as control over the animations. PBL provides learners with a task that closely resembles a real-world problem and allows them to learn by doing. Several studies have been performed on the effectiveness of PBL, but they are frequently related to Science, Technology, Engineering, and Math (STEM) education. The PBL education in the apparel design domain remains to be explored. Using multi-media technologies to develop PBL tasks has been identified as a solution to impart critical and analytical thinking to students [14]. VR can provide a platform to examine and modify prototypes in a real-world setting [15].”

We reminded this rationale to the readers throughout the manuscript, and also added the following summary in the Conclusions section based on your suggestion and highlighted them in the text:

Lines 512-514: “The PBL questions were designed based on relevance to real-world garment design challenges and the need to develop spatial visualization and fit correction skills required for using 3D CAD software programs.”

Lines 524-526: “ The-VR based PBL approach proved effective because it ensured engagement, provided immediate feedback and enabled quick iteration.”

---

## [Decision Letter · Decision Letter 2]

23 Sep 2024

Design and evaluation of a problem-based learning VR module for apparel fit correction training

PONE-D-23-25449R2

Dear Dr. Fatma Baytar,

We’re pleased to inform you that your manuscript has been judged scientifically suitable for publication and will be formally accepted for publication once it meets all outstanding technical requirements.

Kind regards,

Pengpeng Hu

Academic Editor

PLOS ONE

Additional Editor Comments (optional):

Reviewers' comments:

Reviewer's Responses to Questions

**Comments to the Author**

1. If the authors have adequately addressed your comments raised in a previous round of review and you feel that this manuscript is now acceptable for publication, you may indicate that here to bypass the “Comments to the Author” section, enter your conflict of interest statement in the “Confidential to Editor” section, and submit your "Accept" recommendation.

Reviewer #5: All comments have been addressed

Reviewer #6: All comments have been addressed

Reviewer #8: All comments have been addressed

2. Is the manuscript technically sound, and do the data support the conclusions?

Reviewer #5: Yes

Reviewer #6: Yes

Reviewer #8: Yes

3. Has the statistical analysis been performed appropriately and rigorously? 

Reviewer #5: Yes

Reviewer #6: Yes

Reviewer #8: Yes

4. Have the authors made all data underlying the findings in their manuscript fully available?

Reviewer #5: Yes

Reviewer #6: Yes

Reviewer #8: Yes

5. Is the manuscript presented in an intelligible fashion and written in standard English?

Reviewer #5: Yes

Reviewer #6: Yes

Reviewer #8: Yes

6. Review Comments to the Author

Reviewer #5: The new version of the manuscript has been improved accordingly. Thus, the reviewer recommend to publish it in Plos One after careful proofreading.

Reviewer #6: A very interesting paper with clear applications in contemporary garment design and fitting, I look forward to reading the published paper

Reviewer #8: In this revised version, the authors have well answered what I concerned in the last round review.

7. PLOS authors have the option to publish the peer review history of their article (what does this mean?). If published, this will include your full peer review and any attached files.

Reviewer #5: No

Reviewer #6: **Yes: **Simeon Gill

Reviewer #8: No

---

## [Editor Report · Acceptance letter]

7 Nov 2024

PONE-D-23-25449R2 

PLOS ONE

Dear Dr. Baytar, 

I'm pleased to inform you that your manuscript has been deemed suitable for publication in PLOS ONE. Congratulations! Your manuscript is now being handed over to our production team.

Kind regards, 

on behalf of

Dr. Pengpeng Hu 

Academic Editor

PLOS ONE